## Overview Review

global public health; lifestyle; quality of life; mental health; sustainable development

**Corresponding author:**
Luana Scrivano;
Email: luana.scrivano2@unibo.it

# Active mobility and mental health: A scoping review towards a healthier world

Luana Scrivano[1] 🔟, Alessia Tessari[2], Samuele M. Marcora[1] and David N. Manners[1]

[1]Department of Sciences for the Quality of Life, Alma Mater Studiorum, University of Bologna, Bologna, Italy and [2]Department of Psychology "Renzo Canestrari", Alma Mater Studiorum, University of Bologna, Bologna, Italy

## Abstract

Research has proven that engaging in active mobility (AM), namely walking and cycling for transportation, significantly enhances physical activity levels, leading to better physical health. It is still unclear whether AM could also offer any mental health benefits. This scoping review aims to provide a comprehensive understanding of the current knowledge on the relationship between AM and mental health, given its crucial role in public health. The authors searched online databases to isolate primary studies written in English involving an adult sample (16 or over). AM was the exposure factor. Many mental health elements were included as outcomes (depression, anxiety, self-esteem, self-efficacy, stress, psychological and subjective well-being, resilience, loneliness and social support, quality of life, mood, life satisfaction and sleep). The results were organised in a narrative summary per each outcome selected, graphical syntheses and an overview of gaps to be further examined. The authors identified a total of 55 papers as relevant. The results show inconsistency in study designs, definition and operationalisation of the variables, approach and methodologies used. A cross-sectional design was the dominant choice, primarily examining data from national public health surveys. Nonetheless, there has been improvement in outcomes of interests, initially mainly the quality of life and affect. Lately, authors have focused on a broader range of mental health-related factors (such as travel satisfaction). The experimental studies showed promising mental health improvements in those who used active modes more than those who used motorised vehicles. It creates a rationale for further research towards implementing a unified theoretical and methodological framework to study the link between AM and mental health. The ultimate goal is to generate solid conclusions that could support building societies and cities through public health promotion and sustainable strategies, like walking and cycling as a means of transport.

## Impact statement

Walking and cycling are effective modes of transportation that can help achieve various public, medical and societal goals. Scientific evidence has shown that active transportation reduces air pollution, helps increase physical activity and improves physical health. However, mental health has received comparatively less attention in the literature. The global spread of mental health conditions, including depression and anxiety, is a significant challenge for individuals and societies. Moreover, factors like good quality of life, positive self-esteem, sleep quality and social connections contribute to overall well-being. This review provides an extensive overview of the link between active mobility (AM) and mental health outcomes, allowing us to identify findings and research gaps. However, reaching clear-cut conclusions on the impact of AM on mental health is challenging due to the varied terminologies and methodologies used in relevant studies. While promising, this review's results emphasise the importance of a fair and systematic approach to understanding active travel's benefits on mental health. A consistent conceptual and methodological framework is essential to support public health policy changes and encourage healthier and more sustainable lifestyles.

## Introduction

The last few decades have seen a rising interest in active transport modes, namely walking and cycling, among academic researchers and public bodies charged with forming health and transport policies. In studying physical activity as a means of transport, researchers have used numerous synonyms for active mobility (AM), such as *Active Transport, Transportation, Active Commute* and *Commuting, Active Travel* and *Sustainable Transport.* The variety of definitions mirrors the inconsistency and the diversity of the approaches used to study this concept. Nonetheless, these descriptions have in common the idea of physical activity transportation as a way "to get from one place to another" to differentiate AM from leisure-time physical activity, exercise and sport.

The positive consequences of physical activity are well-established in the literature. Regular physical activity is known to have long-term benefits on the population's health. As a non-pharmacology therapy, physical activity does not show drug side effects and is associated with better physical performance and cardiovascular improvements (Marques et al., 2020). Furthermore, physical activity has been reported as a protective factor against mental illness of various natures (Marques et al., 2020).

In particular, public health experts have promoted walking and cycling as they contribute to total physical activity levels across all age groups, improving populations' health and well-being (Kelly et al., 2018). In addition, walking and cycling as active travel modes are of interest from urban and environmental perspectives because of their potential impact on reducing traffic congestion and air pollution (Alattar et al., 2021). Purposeful travel, especially to and from work or educational sites, is a substantial part of people's daily routines. AM could often substitute for journeys undertaken by motorised transport, but policies and interventions are needed to shift the transport paradigm away from using motor vehicles. There has been a considerable amount of research on health promotion, focussed on specific outcomes, such as the relationship between physical activity and overall health and AM and physical health. However, no clear evidence exists of the relationship between AM and Mental Health outcomes, even though mental health problems contribute to the global health burden (Kelly et al., 2018).

Most published evidence refers to *mental health* or *well-being* as umbrella terms, operationalised and measured using instruments that assess elements of behaviour connected to psychological health, where the items investigate mental health components as a total score. However, mental health is a complex concept characterised by a "conglomerate of indicators", including "health determinants, and the severity of symptoms" (Tannenbaum et al., 2009). In the early 2000s, Keyes posited that it was best explained as a combination of emotional, psychological and social well-being (Keyes, 2007). "Well-being" is linked more to psychological than physical health (Galderisi et al., 2015). *Quality of life* is a fundamental aspect of well-being regarding physical and psychological components (De Geus et al., 2008). Consequently, it is an essential element in the public health perspective; indeed, it is one of the most studied mental health-related outcomes regarding active transport.

Beyond the concepts listed above, other aspects of mental well-being that have been evaluated in the context of AM include eudaimonia (Table 1; Ryff, 1989), positive and negative affects, life satisfaction (Gatersleben and Uzzell, 2007), self-efficacy (Bandura, 1977; Pearlin and Schooler, 1978; Rothbaum et al., 1982), negative indicators such as stress (Cohen et al., 1983; Anable and Gatersleben, 2005; Gottholmseder et al., 2009), anxiety (Katsarou et al., 2013), exhaustion (Hansson et al., 2011), sleep quality (Scott et al., 2021) and depression (Kelly et al., 2018).

To evaluate outcomes more closely related to the specific kinds of behaviour of interest, researchers have developed a context-dependent state termed *travel satisfaction* or *commuter satisfaction*, derived from customer satisfaction research (Fornell et al., 1996), which has been linked to more general aspects of mental health (Olsson et al., 2013; De Vos et al., 2022). Numerous factors mediate the relationship between the two levels of analysis: environmental, such as social support (Pearson, 1986; Panter and Jones, 2010; Rovniak et al., 2010; Van Dyck et al., 2011; Paudel et al., 2021), and personal, including mental health "protective" factors like self-efficacy, self-esteem and resilience (Keyes, 2002, 2005, 2007).

Most evidence about the beneficial impact of AM on people's mental health comes from the economic or transport studies fields, not from psychologists. A science journalist recently noted that a "morass of definitions and measurements" is a common problem in social sciences and other scientific fields, quoting Dr Jessica Flake, a quantitative psychologist from McGill University (Gupta, 2022). The findings of the present review demonstrate the presence of this inconsistency in the AM literature. Hence, this scoping review aims to identify evidence of the relationship between active transport modes and mental health outcomes, create a picture of what is already known, what methodologies have been used, and detect research gaps, to identify the areas where research is still required, and proven research methodologies that can evaluate and the beneficial connection between AM and overall healthier life.

The initial research questions were (1) "What methods have been used to investigate the relationship between AM and Mental health?" (2) "What evidence exists in the research literature about the relationship between AM and Mental Health in the general adult population?" (3) "What limitations are apparent in previous studies?"

## Materials and methods

Considering this phenomenon has been addressed only recently and through various approaches, the review was structured as a scoping study "to address broader topics where many different study designs might be applicable" (Arksey and O'Malley, 2005).

### Search strategy

The search strategy aimed to find studies in the research literature investigating the link between walking and cycling as a means of transport and mental health outcomes up to December 2022. Eligibility criteria were primary study; any study design; at least one mental health-related outcome; AM as the exposure; adult sample (18 over) with no specific health condition (e.g., stroke, cancer, postpartum, fibromyalgia and others listed in the literature); articles published in an indexed scientific journal in the English language. The strategy aimed to conduct an exhaustive search of electronic databases. EBSCOhost, PROQUEST, SCOPUS, Web of Science, and PubMed were identified as appropriate topic-related databases and last searched on 30 January 2023; the search had no time limits, any results prior to the date of search were accepted.

Each collection was searched using the following combination of terms:

active mobility OR active travel* OR active transport* OR active commut* OR cycl* OR walk*

AND

mental health OR psychological health OR brain health OR cognitive function* OR depress* OR anxiety OR quality of life OR life satisfaction OR self-esteem OR stress OR psychological well-being OR personal well-being OR subjective well-being OR resilience OR social support OR loneliness.

The search revealed that scientific evidence about AM has primarily focused on young populations and school travelling, especially regarding cognitive functioning. Therefore, the authors added NOT children, NOT adolescents, NOT school to the initially selected terms. The keywords were previously identified following the main objective of this paper, investigating the literature that has previously considered the relationship between AM and mental

**Table 1.** Definitions of the included mental health outcomes

| Mental health outcome | Description |
|---|---|
| Mental health | A state of mental well-being that enables people to cope with the stresses of life, realise their abilities, learn well and work well, and contribute to their community (World Health Organization, 2022) |
| Quality of life | An individual's perception of their position in life in the context of the culture and value system in which they live. This perception concerns their goals, expectations, standards and concerns (Bowling, 2001) |
| Affect (Mood) | Transitory emotional state. The presence of positive emotions and the absence of negative emotions represent the affective component of hedonic well-being (Diener and Emmons, 1984) |
| Eudaimonia | The experience of meaning or purpose, the development of personal strengths, and contribution to society. Also referred to as eudaimonic well-being (McMahan and Estes, 2011) |
| Life satisfaction | Retrospective evaluation of overall happiness and satisfaction, measuring how people feel about their life. The cognitive component of hedonic well-being (Diener et al., 1999) |
| Travel satisfaction | A multi-item measure of how one feels about the travel experience. It comprises a cognitive (quality of travel independently of mode) and two affective components (context-specific factors that stimulate momentary affects) (Friman et al., 2013) |
| Stress | Any change that causes physical, emotional or psychological strain in response to anything that requires attention or action (World Health Organization, 2023) |
| Depressive symptoms | Mood disorder is categorised by prolonged periods of low mood, or lack of interest and/or pleasure in everyday activities, most of the time (American Psychiatric Association, 2013) |
| Anxiety | An emotion characterised by feelings of tension, worried thoughts, and physical changes like increased blood pressure (American Psychological Association, 2022) |
| Loneliness | A subjective unpleasant, or distressing feeling of a lack of connection to other people, along with a desire for more, or more satisfying, social relationships (Position Statement: Addressing Social Isolation and Loneliness and the Power of Human Connection, 2022) |
| Social support | Providing assistance or comfort to others, typically to help them cope with stressors. It may arise from any interpersonal relationship. It may be tangible (material assistance) or emotional (allowing the individual to feel valued, accepted, and understood) (APA Dictionary of Psychology, n.d.) |
| Self-efficacy | The degree of confidence that an individual has in his/her capacity to perform a given behaviour or to overcome barriers (Bandura, 1986) |
| Self-esteem | The extent to which we feel positive or negative about ourselves reflects an individual's subjective evaluation of self-worth and attitudes about the self (Rosenberg, 1965) |
| Resilience | Healthy functioning after a highly adverse event or a conscious effort to continue in an insightful and integrated positive manner as a result of lessons learnt from an adverse experience (Southwick et al., 2014) |
| Vitality/Exhaustion | Physical or intellectual vigour or energy/state of extreme fatigue (Ware and Sherbourne, 1992) |
| Sleep | Individual's self-satisfaction with all aspects of the sleep experience: sleep efficiency, latency, duration, and wake after sleep onset (Nelson et al., 2022) |
| Self-perceived health | Believes about personal health as excellent or poor and likely to get worse (Ware and Sherbourne, 1992) |

health. Relevant studies were screened at a title and abstract level and downloaded on Mendeley software by one of the authors (L.S.). Duplicates were automatically removed. The same author (L.S.) conducted the full-text assessment, and the inclusion of potential studies was agreed upon by consensus with the others (D.N.M., A.T., S.M.M.).

### Data extraction and data management

In line with FAIR data management (FAIR Principles, n.d.) and following PRISMA guidelines (Moher et al., 2009), the authors planned and mutually agreed upon a data extraction and management process. First, they created a chart listing relevant information as follows:

| |
|---|
| Author and year of publication |
| Sample characteristics (number, age, sex, other important sample characteristics) |
| Sampling method (including country/city included) |
| Study design |
| Principal exposure measure (relating to active mobility) |
| Most significant covariate (where applicable) |
| Principal outcome measures (related to mental health) |
| Outcome measures related to physical health (where applicable) |
| Main results |
| Quantified results of outcome measures related to differences in exposure |

One of the authors (L.S.) completed the extraction and was verified by a second author (D.N.M.). Where disagreements arose, these were resolved by consensus. The extracted data were used to describe characteristics and findings for each outcome using narrative summaries. Then, the authors identified gaps in the literature and outlined areas that need further research. According to an Open Science framework, all data and metadata are available online within Supplementary Appendices A and B.

### Study quality and risk of bias

Study quality was assessed mainly in terms of study design. In terms of the Oxford Centre for Evidence-Based Medicine classification (OCEBM Levels of Evidence Working Group, 2011), all studies included in the review were in the range of step 2 ("randomised trial" (s)), step 3 ("non-randomised controlled cohort/follow-up studies") or step 4 ("Case-series or case–control studies"). Studies using designs at a higher step are more at risk of generating findings biased by confounding variables than those at a lower step.

The OCEBM Levels of Evidence Table was used to assign the designation of each study, without taking into account effect size or systematically assessing study quality. The consensus process used was the same as that employed in the search phase (section "Search strategy").

### Graphical syntheses

The first step in synthesising results after tabulating study characteristics was to summarise data items graphically to facilitate a qualitative comparison of all the studies and identify relevant points of similarity or difference. Next, extracted data were processed using scripts developed in the R programming language.

#### Word cloud

The most commonly occurring two-word expressions associated with the studies summarised were determined as follows. First, the pdf files of the studies were cleaned to remove punctuation and generic commonly occurring words. Then two-word expressions were extracted and counted, merging entries that differed only for capitalisation (Benoit et al., 2018). The 500 most commonly occurring two-word expressions were manually reviewed by two authors and sorted as being related to exposures, covariates, outcomes, or other aspects of the studies. Word clouds were created using version 0.2.2 of the Wordcloud2 package, with entries related to the frequency of occurrence, to provide an objective presentation of the themes contained within the studies.

#### Geographical mapping

The distribution of sampled populations was projected onto a world map, using symbols proportional to the sample size. Extracted locations were identified using geocoding based on the cities or regions named as sample locations or the geographical centre of any country from which a national sample was obtained.

#### Albatross plot

The present review does not aim to assess the weight of evidence but to map the claimed results so that the reader can rapidly assess the different studies. Using the Albatross plot (Harrison et al., 2017), an established method in line with this aim, four plots were constructed through the Metap package (version 1.8). Based on the $p$-value of the statistical test undertaken (recorded in the data extraction table) and the sample size used to provide the data, an implicit effect size can be calculated. Statistical tests were included if they related to a difference between the use of active transport modalities and other forms of transport (taking car use as the baseline comparison where possible, or public transport otherwise). However, given the various tests used in the different studies and differences in reporting the $p$-value, this should be interpreted only as a rough guide as to which studies reported which outcomes and whether these were small or large, beneficial or harmful.

### Descriptive analytical synthesis

Data were organised using the list of mental-health-related outcomes in Table 1, congruent with the wordcloud described above. Then, we mapped the instruments and noticed that outcomes were given different names but measured with the same instrument or vice versa. Considering the variability of exposures, outcomes and methodology approach used in the included literature, the authors outlined the findings per each outcome following the narrative tradition approach of descriptive-analytical method, which consists of applying a standard analytical framework to each study and collecting specific information reported (Arksey and O'Malley, 2005), as in section "Data extraction and data management".

## Results

### Search results

Figure 1 presents the flow diagram of the study selection. Database searches identified 502 records. After duplicates were excluded, each paper was assessed at a title/abstract level and, if selected, full-text level. Studies were excluded when the outcome or exposure did not align with the research questions. A total of 55 relevant studies were retained.

### Study characteristics

Study characteristics are summarised in Table 2. The majority (40) employed a cross-sectional design based mainly on data collected by public health surveys, typically including subjects who could autonomously travel using passive or active travel modes. Nine studies reported a longitudinal design; however, data were mainly analysed following a cross-sectional approach. Lastly, two studies had a quasi-experimental design in which the sample was assigned to intervention or control groups following deterministic criteria. We found only one unrandomized controlled trial and four randomised controlled trials, all investigating changes in the quality of life from 2000 to 2020. There is no evidence of an evolution in study designs; the oldest study (Mutrie et al., 2000) had an experimental design (RCT), while the most recent (Scarabottolo et al., 2022) used a cross-sectional analysis. On the other hand, outcomes of interest have developed. Before 2010, studies mainly focused on outcomes like Quality of Life and Affect, while more recently, the focus has shifted to other potentially important aspects. Demographically, most study participants were women. Given our selection criteria, the study populations ranged from 15 to 98 years old. Some of the included studies specifically focused on older adult samples (60–98 years old) where AM ("outdoor mobility") is considered essential to maintaining good quality of life levels.

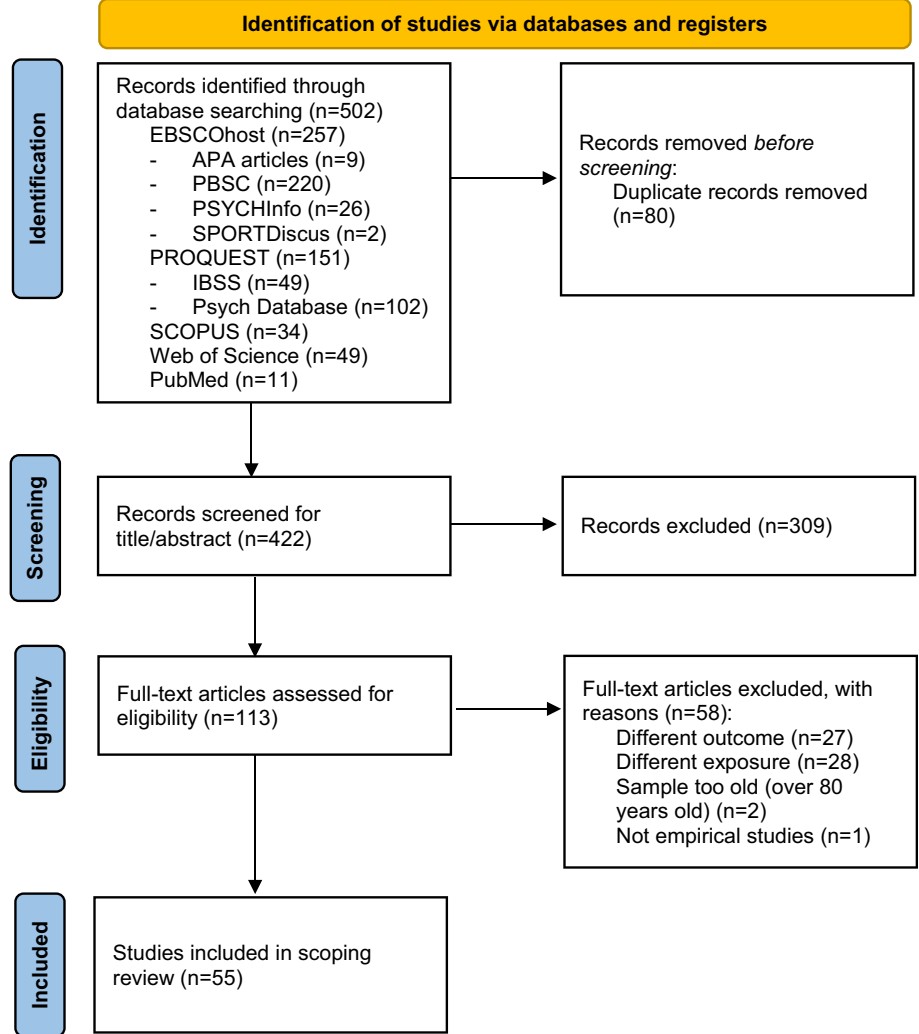

**Figure 1.** PRISMA 2020 flow diagram for new systematic reviews which included searches of databases and registers only.

## Graphical mapping of studies

Firstly, word clouds of phrases related to outcomes and covariates are shown in Figure 2. They were used to validate the concepts attended to in the data summary phase and are referenced at appropriate points in the text below. Secondly, the distribution of studies is illustrated on a world map in Figure 3. Except for some Chinese and Brazilian studies, most work in this field is concentrated in Europe and the English-speaking world. Necessarily, the largest studies are cross-sectional, primarily based on data from national surveys, not purposively collected to evaluate either mental health, active transport or the relationship between these. Finally, the implicit effect sizes of statistically assessed outcomes are shown in Figure 4, dividing the literature into four quadrants, since the high number of mental health outcomes is considered. It is immediately apparent that most studies are compatible with a hypothesis of moderate benefit due to AM or no effect. Only one finding is of a large dis-benefit.

## Mapping of terminology

This section examines how AM as a causal factor and mental health as an outcome have been described and categorised.

## AM, the exposure factor

While studies have shown that AM improves physical health (Alattar et al., 2021), it is not always the case for mental health. The current scoping review is part of a larger project examining the effects of AM on different aspects of well-being in the adult population. As such, it focused on studies that considered walking and cycling as a means of transport, travelling, and commuting, in short, "to get from one place to another" (Physical Activity Guidelines Advisory Committee, 2018), and not for exercise, sport or leisure purposes. As such, authors approved "AM" as an advantageous nomenclature since it communicates the action of "moving actively" without excluding any transport purpose (e.g., "active commuting" seems to specifically refer to travels from home to work/school or vice versa). Also, AM is commonly used in Europe, so it might reduce language and understanding biases in our future studies, taking place in Italy. The research literature refers to active travel primarily in terms of walking and cycling, while public transport is often investigated separately and is usually related to comparatively adverse health results (Office for National Statistics, 2014; Neumeier et al., 2020; Jacob et al., 2021), but partially overlapping terms are abundant. Marttila and Nupponen (2000) defined *everyday commuting activity (ECA)* as one of two Health Enhancing Physical Activity (HEPA) categories. At the same time,

**Table 2.** Summary of study characteristics and participants, ordered by study design and year of publication

| (A) Non-experimental studies | | | | | | |
|---|---|---|---|---|---|---|
| References | Original study/ Project | Country (city/region) | Sample size (ages) % women | Participant characteristics | Code | Active mobility measure(s) |
| *Cross-sectional studies* | | | | | | |
| Gatersleben and Uzzell (2007) | Stand-alone | UK (Surrey) | 389 (19–64) 51% | University employees | M | F |
| Ohta et al. (2007) | Stand-alone | Japan (Kita Kyushu) | 670 (39.9) 36% | Municipal employees | C W | Do you walk or cycle to work? + D |
| Cerin et al. (2009) | PLACE (Physical activity in Locality and Community Environments) | Australia (Adelaide) | 2,194 (20–65) 64% | | C W | IPAQ |
| Jurakić et al. (2010) | Stand-alone | Croatia | 1,032 (15–199) 52% | | C W | IPAQ |
| Rasciute and Downward (2010) | Taking Part Survey – British Market Research Bureau | England | ND (16–199) | | C W | F + D |
| Bergland et al. (2010) | NorLAG (Norwegian Life Course, Ageing and Generation Study) | Norway | 3,069 (55–79) 51% | Older adults living in their own homes ("in the community" or "community-dwelling adults") | W | Di |
| Molina-García et al. (2010) | Stand-alone | Spain (Valencia) | 518 (22.4) 60% | Students from two universities | M | F |
| Hansson et al. (2011) | Scania Regional Health survey | Sweden (Scania) | 23,111 (18–65) 50% | Working >30 h/week | M | D |
| Smith (2012) | Stand-alone | USA | 12 (74–98) 67% | Older adults living in their own homes ("in the community" or "community-dwelling adults") | M | Daily Life Interview |
| Pucci et al. (2012) | "Level of PA among adults: association with perceived environment and social support" project | Brazil (Curitiba) | 1,461 (40–59) 63.7% | Individuals living around green areas (parks, squares) | W | IPAQ |
| Gómez et al. (2013) | Stand-alone | Colombia (Cali) | 1,263 (18–59) 100% | Women living in low and middle-low socioeconomic-status areas | W | IPAQ |
| Humphreys et al. (2013) | Cambridge Study | UK (Cambridge) | 989 (16–199) 68% | Working adults | C W | F + D |
| ONS (2014) | Annual Population Survey (APS) | UK | 60,200 () | Commuters and non-CtW | M | D |
| St-Louis et al. (2014) | McGill University-wide commuter survey | Canada (Montréal) | 3,377 (30.43) | University staff and students | M | D |
| Morris and Guerra (2015) | American Time Use Survey (ATUS) | USA | 13,269 (15–199) | | M | M |
| Mason et al. (2016) | GoWell Research and Learning Programme | Scotland (Glasgow) | 2,654 (18–199) 60% | Residents of most deprived neighbourhoods | C W | F |
| Bélanger-Gravel et al. (2016) | BIXI (Bicycle-taXI; Public Bicycle Share Program) | Canada (Montréal) | 3,978 (18–65) 60% | Users of bicycle sharing scheme | C | "Have you ever used PBSP?" |
| Chng et al. (2016) | UKHLS (UK Household Longitudinal Study) | UK (London) | 3,630 (16–65) 53% | | M | M |
| Friman et al. (2017) | Stand-alone | Sweden (Stockholm, Göteborg, Karlstad) | 367 (18–199) 63% | CtW | M | F |

*(Continued)*

**Table 2.** (*Continued*)

| (A) Non-experimental studies | | | | | | |
|---|---|---|---|---|---|---|
| References | Original study/ Project | Country (city/region) | Sample size (ages) % women | Participant characteristics | Code | Active mobility measure(s) |
| Sener and Lee (2017) | El Paso Regional Multimodal Transportation Survey | USA (Texas) | 1,505 (18–199) 53% | | M | F |
| Smith (2017) | Stand-alone | USA (Oregon) | 828 (25–44) 45% | CtW | M | D + Di |
| Avila-Palencia et al. (2017) | TAPAS (Transportation, Air Pollution and Physical Activities) | Spain (Barcelona) | 788 (18–199) 52% | CtW | C | F |
| Lancée et al. (2017) | "Geluks Wijzer" (Happiness Indicator) study | Netherlands | 1,450 (15–71) 82% | CtW | M | M + Time of the day |
| Mattisson et al. (2018) | Stand-alone | Sweden (Scania) | 7,574 (18–65) 55% | Working 15–60 h/week | M | M |
| McCarthy and Habib (2018) | Nova Scotia Travel Activity (NovaTRAC) survey | Canada (Nova Scotia) | 493 (41.1) | | M | F + D + Di |
| Foley et al. (2018) | UK Harmonised European Time Use Survey (2014/15) | UK | 6143 (48) 53% | | C W | M |
| Vancampfort et al. (2018) | SAGE (Study on Global Ageing and Adult Health) study – WHO | China, Ghana, India, Mexico, Russia, South Africa | 14,585 (65–199) 55% | Older adults living in their own homes ("in the community" or "community-dwelling adults") | C W | GPAQ + F + D |
| Brainard et al. (2019) | ALS1617 (Adult Active Lives Survey 2016/2017) | England | 51,780 (16–104) | | C W | F + D |
| Scarabottolo et al. (2019) | Stand-alone | Brazil (Presidente Prudente) | 400 (60–199) 58% | Older adults living in their own homes ("in the community" or "community-dwelling adults") | C W | Baecke Questionnaire |
| Kaplan et al. (2019) | Stand-alone | Australia (Brisbane) | 1,131 (17–199) 47% | | C | F |
| Singleton (2019) | Positive Utility of Travel (PUT) study | USA (Oregon) | 682 (18–199) 55% | CtW | C W | D |
| Clark et al. (2020) | Understanding Society survey (previously UKHLS and prior BHPS) | UK | 26,551 () | Working adults | M | D |
| Sattler et al. (2020) | Health On The Way (HOTway) | Austria (Graz) | 188 (18–64) 49% | University employees or students | M | M |
| Fan et al. (2021) | Stand-alone | China (Beijing) | 1,080 (18–199) 49% | | M | M+ D+ Purpose+ time of day + day of week + Perceived trip D + Companion + Activities (during travel) |
| Lira and Paez (2021) | Larger Survey in Santiago (Chile) | Chile (Santiago) | 451 (18–199) | | M | M |
| Liu et al. (2021) | Stand-alone | China (Heze) | 188 (18–199) | | M | M |
| Muñiz et al. (2021) | European Health Survey in Spain (EHSS – 2014) | Spain | 16,121 (15–199) 53% | | C W | F |
| Wang et al. (2021) | Stand-alone | China | 16,103 (47) 52% | Working adults | M | D |
| Cobbold et al. (2022) | STASH (Sydney Travel and Health Study) | Australia (Sydney) | 532 (18–55) 60% | | M | F + D |

**Table 2.** (*Continued*)

| (A) Non-experimental studies | | | | | | |
|---|---|---|---|---|---|---|
| References | Original study/ Project | Country (city/region) | Sample size (ages) % women | Participant characteristics | Code | Active mobility measure(s) |
| *Longitudinal studies* | | | | | | |
| Lampinen et al. (2006) | Evergreen Project | Finland (Jyvaskyla) | 663 (65–84) 67% | Elderly | W | Di |
| Martin et al. (2014) | BHPS (British Household Panel Survey) | UK | 17,985 (18–65) 49% | CtW | M | M |
| Mytton et al. (2016) | Cambridge Study | UK (Cambridge) | 801 (16–199) 69% | CtW | C W | F + D |
| Avila-Palencia et al. (2018) | PASTA (Physical Activity through Sustainable Transportation Approaches) | Spain (Barcelona) | 3,567 (18–199) 53% | | M | F |
| Knott et al. (2018) | Stand-alone | UK (Stockport) | 5,474 (40–75) 46% | CtW | M | F + Di + M |
| Glasgow et al. (2019) | Stand-alone | USA (VA, DC, MN) | 229 (18–65) 55% | Owners of Android smartphone | M | M + F + D + Purpose +Activities |
| Yang et al. (2019) | Stand-alone | USA | 18,400 (65–85) 50% | Older adults living in their own homes ("in the community" or "community-dwelling adults") | M | D |
| Kroesen and De Vos (2020) | Longitudinal Internet Studies for the Social Sciences panel (LISS) | Netherlands | 1,548 (15–199) 49% | Stratified population survey sample | W | F |
| Scarabottolo et al. (2022) | Stand-alone | Brazil (Presidente Prudente) | 331 (40–80) 68% | | C W | Baecke Questionnaire |

| (B) Quasi-experimental and experimental studies | | | | | | | |
|---|---|---|---|---|---|---|---|
| References | Design | Original study/Project | Country (city/ region) | Sample size (ages) % women | Participant characteristics | Code | AM measure(s) |
| de Geus et al. (2008) | Non-randomised CT | Stand-alone | Belgium (Flanders) | 80 (30–65) | Members of an insurance company | C | F+ D+ Di |
| Page and Nilsson (2017) | Quasi-experimental | Stand-alone | UK | 31 (21–55) 80% | Employees of global education provider | E | F + D |
| Jacob et al. (2021) | Quasi-experimental | UKHLS (UK Household Longitudinal Study) | England | 31,736 (16–65) 59% | Stratified population survey sample | M | M |
| Mutrie et al. (2000) | RCT | Stand-alone | Scotland (Glasgow) | 237 (19–69) 63% | CtW (university, hospital trust, and district health board) | C W | SPAQ |
| Mutrie et al. (2002) | RCT | "Walk in to Work out" project | Scotland (Glasgow) | 295 (16–69) 64% | CtW (university, hospital trust, and district health board) | C W | SPAQ |
| Baker et al. (2008) | RCT | WWW (Walking for Well-bing in the West) | Scotland (Glasgow) | 79 (18–65) 80% | Sedentary adults in the low socioeconomic group | W | IPAQ+F + D |
| Neumeier et al. (2020) | RCT | GISMO (Geographical Information Support for healthy Mobility) project | Austria (Salzburg) | 62 (37–55) 62% | CtW (hospital employees) | C P | Di + D+ F |

*Note:* Ages presented as range when available or otherwise average where available. Design – RCT, randomised controlled trial. Participant characteristics – CtW, commuters to work. Coded mobility mode – C, cycling; E, e-biking; M, multi-mode; P, public transport; W, walking. Coded active mobility measure – D, duration; Di, distance; F, frequency; IPAQ, international physical activity questionnaire; M, mode; SPAQ, Scottish physical activity questionnaire.

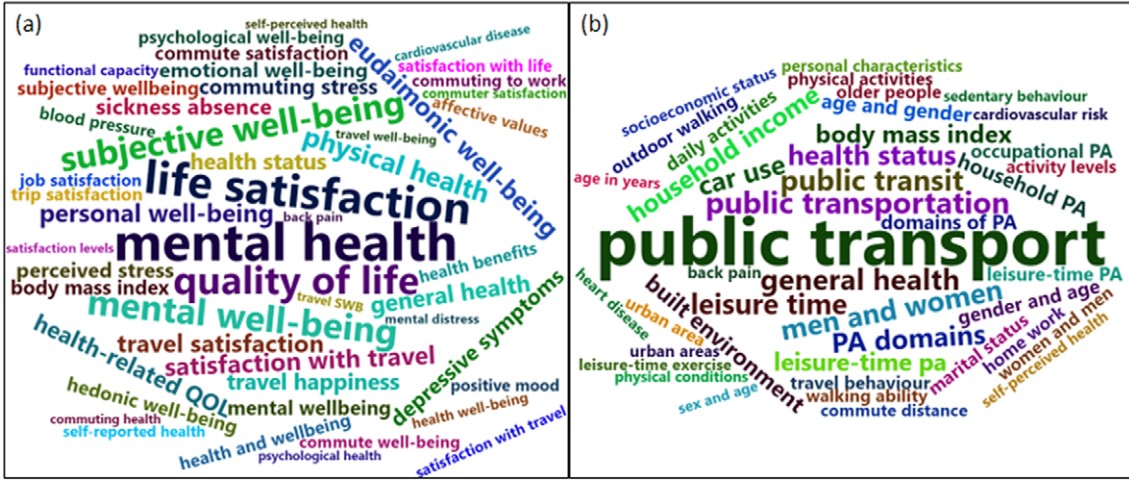

**Figure 2.** Word clouds summarising the most common study-specific phrases found in articles included in the review are categorised as (a) related to outcomes or (b) related to covariates. The size of the text is proportional to the relative number of occurrences of the phrase.

Mutrie et al. (2000, 2002) implemented two RCTs to study active commuting behaviour and trips explicitly to work or educational sites. Outside the UK, the term "active commuting" has been used worldwide, for example, in Japan (Ohta et al., 2007), New Zealand (Tin Tin et al., 2009), Sweden (Hansson et al., 2011), Canada (St-Louis et al., 2014) and China (Liu et al., 2021). While *active commuting* appears to be limited to trips whose purpose is primarily the journey to/from work (or university), authors have argued that walking and cycling have several utilitarian transportation purposes, and going to work is just one of them (Singleton, 2018). Terms including active *travel, travelling, transport* or *transportation* suggest that the end of the movement might be to complete other everyday tasks outside the proper "exercise" and "leisure-time" objectives, such as going to the supermarket or the post office. In the UK, the iConnect Study (Impact of Constructing Non-motorised Networks and Evaluating Changes in Travel) (Ogilvie et al., 2010, 2011) was a natural experiment that evaluated the effect of *active travel*, "walking and cycling as travel behaviour" on people's health, traffic congestion, and carbon emission. Later, the USA Physical Activity Guidelines Advisory Committee (2018) reported that transportation physical activity examples include "walking or bicycling to and from work, school, transportation hubs, or a shopping centre". Finally, Alattar et al. (2021) defined *active travel* as "journeys that have been undertaken either entirely or partially using human-powered transportation modes such as walking, cycling or using a wheelchair". More recently, European researchers involved in the PASTA (Physical Activity through Sustainable Transport Approaches) project defined activities, including "walking and cycling for transport solely or in combination with public transport", as AM (Gerike et al., 2016). As Redding et al. (2014) stated, walking and cycling can be classified as *alternative/sustainable transportation* "defined as commuting by any means other than a single occupancy vehicle (SOV)".

### Mental health, the outcome factor(s)

In the present study, the authors looked for studies that examined outcomes related to mental, cognitive, neurological and brain health potentially related to AM. Considering the iterative process entailed in a scoping review, we were open to any possible results and ready to collect various factors offered and change the research process accordingly. The wide range of possible outcome measures is evident in Figure 2 on the states and symptoms mentioned repeatedly in the studies initially identified; we selected 17 mental health-related outcomes (Table 1) and examined how these were defined and operationalised.

Most published evidence refers to *mental health or well-being* as umbrella terms, operationalised and measured using instruments that assess elements of behaviour connected to psychological health, where the items investigate mental health components as a total score or divided between various domains. While mental health refers to a state, "quality of life" is related to individuals' perceptions and includes physical and psychological components (De Geus et al., 2008). Both are essential components in the public health perspective and furnish necessary outcome measures in active transport studies, although the distinctions are not always clearly defined.

Well-being was another frequently used term, and we decided to treat each component of well-being separately (eudaimonia, affect, life satisfaction), avoiding using general terms like subjective, psychological or personal well-being that could lead to further confusion. We found mainly cross-sectional designs investigating if travelling using physical activity could make people happier and improve their mood or affect. Therefore, we decided to unify these studies under one term, *affect*, regarding positive and negative affective aspects of well-being. Compared to affective factors, the concept of *Eudaimonic Well-Being* is less well-represented in the studies reviewed. Eudaimonia refers to the sense of worthiness in people's lives, not only their mood (Office for National Statistics, 2014). Only recently have researchers considered that commuting might also impact the cognitive component of subjective well-being, *life satisfaction*.

While searching for this outcome, authors noted the term *travel satisfaction*, which was not included initially. Also called "commute well-being", it is defined as a "multi-item measure of how one feels about the commute to work and its associated factors" (Smith, 2017), which is a cognitive evaluation of the quality of travel and an affective evaluation of feelings during travel (from stressed to relaxed and from bored to excited) (Ettema et al., 2011). In line with subjective well-being, travel satisfaction was included in the present review because it represents aspects of personal well-being specifically involved in commuting (Ettema et al., 2010) and directly correlates with life satisfaction and affect (Friman et al., 2017) and indirectly with eudaimonic well-being (Liu et al., 2021).

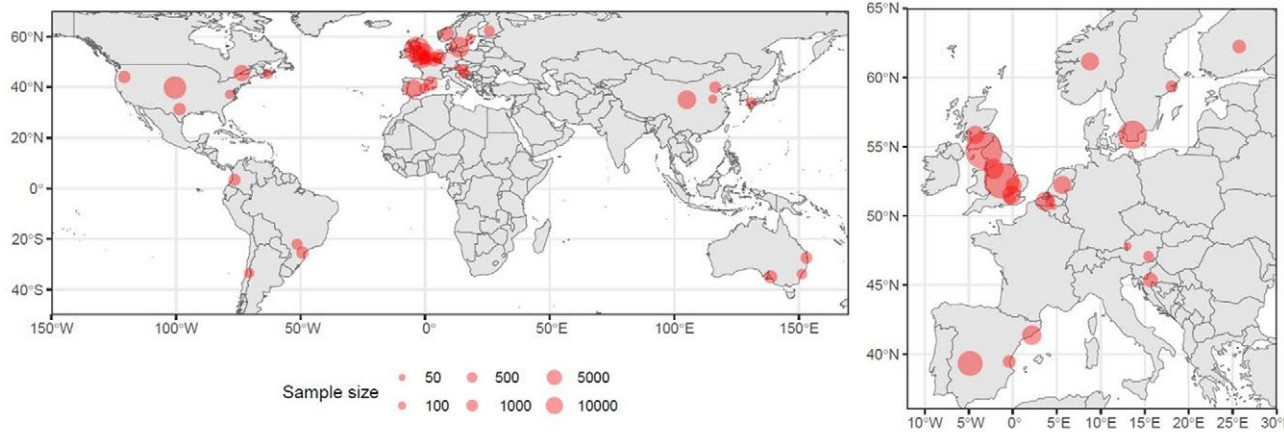

**Figure 3.** The study distribution in the review is projected onto the world map (left panel), while European studies are shown in the enlargement (right panel). The size of the symbols is related to the number of participants. The location of each symbol is centred on the centre of the city, region or country from which participants were drawn.

Due to its favourable influence on mental and physical health (Pearson, 1986), *social support* was included among the original search terms. Meanwhile, we encountered evidence of scientific interest in active commuting and *loneliness.* It regards a subjective state, distinct from the objective condition of not having social contact, called "social isolation" (Masi et al., 2011). Negative, distressing emotions are also associated with *loneliness* (de Jong Gierveld and van Tilburg, 2016). The literature has described the crucial role of social support in promoting or discouraging physical activity. Hence, the paper included social support and loneliness as outcomes of mental health and public health interest.

Similarly, previous research indicated *self-efficacy* as a considerable correlate for physical activity, more as a motivator than a benefit. Nonetheless, we wanted to emphasise this concept as an essential health outcome. Therefore, according to the definition and operationalisation of this term (Bandura, 1977), we merged the references about "mastery, confidence, achieving goals" and other subscales of self-efficacy (e.g., physical self-efficacy) into one outcome.

While screening the selected literature, Vitality became more susceptible to the benefits of AM. Experimental results reported that improved walking scores in the intervention group also improved quality of life scores, specifically in the Vitality subscale (Mutrie et al., 2002; De Geus et al., 2008). A longitudinal design reached the same result (Scarabottolo et al., 2022). We noticed some literature considering this one quality-of-life domain as the outcome. Still, studies measuring "low levels of energy or vitality" used the title *exhaustion*, even though the instrument employed was the same (SF-36 Vitality subscale). We decided to reserve specific attention to this item.

We encountered the term *self-reported health* during the screening phase, also mentioned as self-perceived or self-rated health, or overall health status or physical health. It seems reasonable that active travel research on mental health focused on this outcome since it reciprocally relates to subjective well-being (Rasciute and Downward, 2010; Clark et al., 2020), and it is also a "valid indicator of morbidity and mortality" (Wilson et al., 2007). Furthermore, other authors have measured "health status" using self-report instruments like the physical component score of the SF-12 (Jacob et al., 2021), given that people might interpret this term as only related to physical health (Clark et al., 2020). Consequently, self-reported health is a likely mediating factor when designing

interventions (Mattisson et al., 2018). Accordingly, we introduced it in this review.

Three of the original search terms remain to be considered. First, although *sleep* patterns are crucial intermediates in the relationship between physical and mental health (Scott et al., 2021), we could only find two studies focusing on sleep quality and active travel. Secondly, we found no study investigating a potential relationship between AM and self-esteem despite being associated with health-related behaviours and contributing to people's health status (Bauman, 2012). In line with our results, a review (Kelly et al., 2018) has recently concluded that the evidence about active travelling improving or maintaining good self-esteem is mixed or inconsistent. The authors found no experimental evidence, and only one (Bergland et al., 2010) of the 11 identified studies examined outdoor walking mobility specifically, while in others, the exposure factor was physical activity or exercise. Finally, we found no evidence of studies on *resilience* and walking or cycling as transport means. However, the review mentioned above (Kelly et al., 2018) claimed that it is an outcome to attention considering the emergent evidence of physical activity in healthy adults (Childs and de Wit, 2014). Therefore, we do not further consider results concerning self-esteem and resilience.

While we aimed to investigate the potential health benefits of active travel modes on the general healthy adult population, most studies about *cognitive health* involved youth samples (children, adolescents, and students under 18) or older adults with cognitive functional impairments, so we excluded studies solely concerned with this outcome. Additionally, we found only one study specifically about *brain health* (Bos et al., 2011), which measured the increase in BDNF serum levels due to active travel moderated by air pollution or the natural environment. Because of the lack of congruence with the goal of this review, we did not include that study either.

*Moderating variables*
Several variables are known to moderate the relationship between AM and its potential health benefits. For example, travel time and environmental characteristics moderate this link (Office for National Statistics, 2014; Clark et al., 2020; Wang et al., 2021). Many of the studies reviewed evaluated one or more of these factors, as evidenced by Figure 2, but limitations in study design preclude the possibility of studying all of them together.

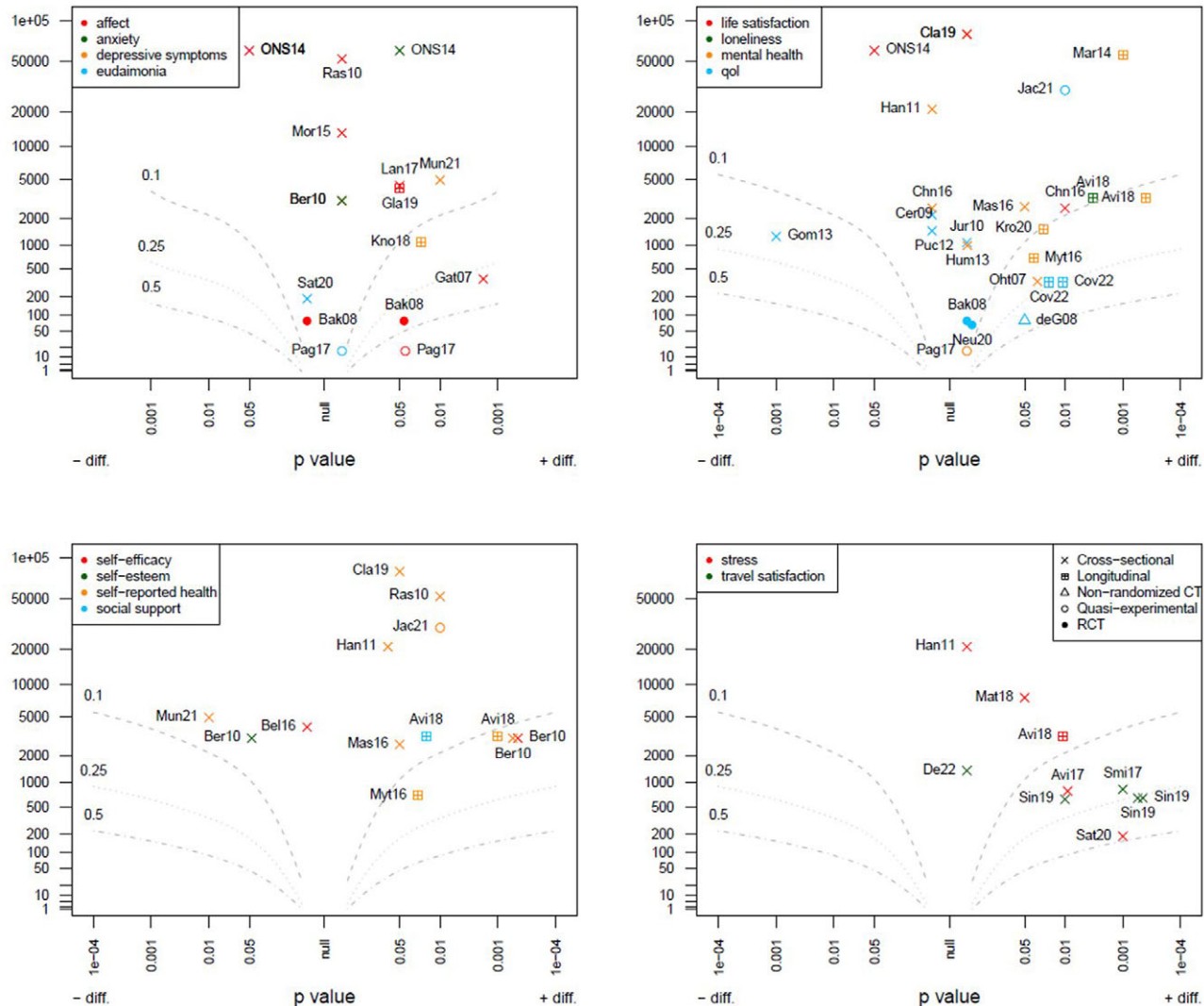

**Figure 4.** Albatross plots mapping the imputed effect size of each study for which data were available regarding differences in outcome between study participants selecting active and motorised mobility for transport needs in daily life. Curved lines join points with equal effect sizes (labelled 0.1, 0.25, 0.5). The *x*-axis indicates the reported *p*-value. Points to the right of the origin (null *p*-value) show a benefit for active mobility, and those to the left show a benefit for motorised transport. The *y*-axis shows the number of participants on a logarithmic scale. Points are positioned conservatively when only the relationship to a significance threshold is reported (i.e., *p* < 0.05 is treated as *p* = 0.05). Results are divided between four panels for easier visualisation, grouping outcomes by alphabetical order. Symbol shapes indicate study design (legend in the bottom right-hand panel). Labels indicate the first three letters of the first author's name and the 2-digit year of publication.

## Mapping of outcomes' instruments and results

This section examines how the above outcomes have been operationally defined and measured. Also, we provide a verbal summary of the study results for each outcome in Table 3. Extended details about each study are also provided in Supplementary Appendix A.

### Mental health and mental well-being

Numerous studies conducted since 2006 have explored the connection between active transportation and overall mental health. The findings are presented in Figure 4. The tools primarily used for *mental health* were the General Health Questionnaire (Goldberg, 1972), validated to assess elements of behaviour connected to psychological health, such as sleep, depression, confidence and focus, the Mental Component Sore from the SF-12 (MCS-SF) (Ware et al., 1996), and the Mental Health Inventory-5 (MHI-5) (Berwick et al., 1991), which includes dimensions of anxiety,

depression, loss of behavioural/emotional control, and psychological well-being. Three longitudinal studies showed a positive association between these variables (Lampinen et al., 2006; Martin et al., 2014; Kroesen and De Vos, 2020), while two others limited the positive association to cycling, not walking (Mytton et al., 2016; Avila-Palencia et al., 2018). However, only two cross-sectional papers demonstrated a positive association (one only for men) (Ohta et al., 2007; Mason et al., 2016), and four did not report any relationship (Hansson et al., 2011; Humphreys et al., 2013; Chng et al., 2016; Clark et al., 2020). Recently, two quasi-experimental studies using different instruments (GHQ-12 and MCS-SF12) showed a positive association between cycling as a transport (Page and Nilsson, 2017) or active travel modes (Jacob et al., 2021) and mental health scores. Previous research has identified the "healthy commuter effect" (Hansson et al., 2011) as a selection bias whereby unhealthy people are less keen on commuting actively. However, one later study (Kroesen and De Vos, 2020)

**Table 3.** Summary of findings relating to the effect of active mobility on aspects of mental health, ordered by mental health variable, study design, and year of publication

| References | Study design | Mental health measure | Evidence of relationship | Largest covariate |
|---|---|---|---|---|
| *Mental health variable: affect* | | | | |
| Gatersleben and Uzzell (2007) | CS | Russell and Lanius (1984) − 7 items | Yes+ | Travel distance |
| Rasciute and Downward (2010) | CS | "Taking all things together, how happy would you say you are?" | Yes+ (W) | Any sport |
| ONS (2014) | CS | "Overall, how happy did you feel yesterday?" | Yes- | Many inc. age, sex |
| Morris and Guerra (2015) | CS | Affect Balance Scale (happiness, sadness, tiredness, pain and stress) | No | Interacting with another person |
| Lancée et al. (2017) | CS | How happy did you feel yesterday? | Yes+ | Duration >60 min (−ve), with someone (+ve) |
| Brainard et al. (2019) | CS | "On a scale of 0–10, where 0 is not at all happy (anxious)and 10 is completely happy (anxious), overall, how happy (anxious) did you feel yesterday?" | No | |
| Kaplan et al. (2019) | CS | Profile Mood State Scale | Yes+ (C) | |
| Lira and Paez (2021) | CS | "Please indicate the mode(s) of transport that you relate to the following feelings: freedom, enjoyment, happiness, poverty, luxury, status" | Yes+ | |
| Glasgow et al. (2019) | Longitudinal | Travel Mood Scale | Yes+ | Destination walk score |
| Page and Nilsson (2017) | Quasi-experimental | Diary (e.g., tired, energised) | Yes+ (C) | |
| Baker et al. (2008) | RCT | PANAS | Yes+ (W) | None |
| Singleton (2019) | CS | Travel Affect | Yes+ | Travel usefulness |
| *Mental health variable: anxiety* | | | | |
| Bergland et al. (2010) | CS | HSCL | Yes+ (W) | Age |
| ONS (2014) | CS | "Overall, how anxious did you feel yesterday?" | Yes+ | Many inc. age, sex |
| Brainard et al. (2019) | CS | "On a scale of 0–10, where 0 is not at all anxious and 10 is completely happy (anxious), overall, how happy (anxious) did you feel yesterday?" | No | |
| *Mental health variable: depressive symptoms* | | | | |
| Bergland et al. (2010) | CS | CES-D | No (W) | Age |
| Muñiz et al. (2021) | CS | Diagnosis/symptoms during the past 12 months | Yes- | Age |
| Knott et al. (2018) | Longitudinal | PHQ-2 | Yes- | NA |
| Yang et al. (2019) | Longitudinal | CESD-8 | No | |
| *Mental health variable: eudaimonia* | | | | |
| ONS (2014) | CS | "Overall, to what extent do you feel the things you do in your life are worthwhile?" | Yes- | Many inc. age, sex |
| Brainard et al. (2019) | CS | "On a scale of 0–10, where 0 is not at all worthwhile and 10 is completely worthwhile, overall, to what extent do you feel the things you do in your life are worthwhile?" | No | |
| Kaplan et al. (2019) | CS | Fleming and Courtney's scale | Yes+ (C) | |
| Liu et al. (2021) | CS | Flourishing Scale | No | No |
| Page and Nilsson (2017) | Quasi-experimental | Flourishing Scale | No (C) | None |
| Singleton (2019) | CS | Travel Eudaimonia | Yes+ | For example, Travel usefulness versus health |
| *Mental health variable: life satisfaction* | | | | |
| ONS (2014) | CS | "Overall, how satisfied are you with your life nowadays?" | Yes- | Many inc. age, sex |
| Chng et al. (2016) | CS | "How dissatisfied or satisfied are you with your life overall?" | Yes+ (W) | NA |

**Table 3.** (*Continued*)

| References | Study design | Mental health measure | Evidence of relationship | Largest covariate |
|---|---|---|---|---|
| Sener and Lee (2017) | CS | 1-item How satisfied are you with your life? | Yes+ (1) | |
| McCarthy and Habib (2018) | CS | "How would you describe your usual attitudes towards life?" | Yes+ | |
| Brainard et al. (2019) | CS | "On a scale of 0–10, where 0 is not at all satisfied and 10 is completely satisfied, overall, how satisfied are you with your life nowadays?" | No | |
| Clark et al. (2020) | CS | "How would you describe your usual attitude towards life?" | No | |
| Wang et al. (2021) | CS | "How satisfied are you with your life on the whole?" | Yes+ | |
| Jacob et al. (2021) | Quasi-experimental | "How satisfied do you feel with your overall life?" | Yes+ | NA |
| *Mental health variable: loneliness* | | | | |
| Smith (2012) | CS | UCLA-Loneliness scale | Yes- | |
| Avila-Palencia et al. (2018) | Longitudinal | UCLA-Loneliness scale | Yes- (C) | |
| *Mental health variable: mental health/mental well-being* | | | | |
| Ohta et al. (2007) | CS | GHQ-28 | Yes+ (M) | Any leisure exercise >3 h/wk |
| Hansson et al. (2011) | CS | GHQ-12 | No | NA |
| Humphreys et al. (2013) | CS | MCS-SF8 | No | Age, total recreational activity |
| Mason et al. (2016) | CS | MCS-SF12+ WEMWBS | Yes+ | Long-standing illness |
| Chng et al. (2016) | CS | GHQ-12 | No | NA |
| Clark et al. (2020) | CS | GHQ-12 | No | Commute time |
| Lampinen et al. (2006) | Longitudinal | RBDI + "Do you think you are lonely?" + "How would you describe your self-rated mental vigour at the moment?" + "Right now, how meaningful do you consider your life?" | Yes+ | NA |
| Martin et al. (2014) | Longitudinal | GHQ-12 | Yes+ | NA |
| Mytton et al. (2016) | Longitudinal | MCS-SF8 | Yes+ (C) | Physical limitation |
| Avila-Palencia et al. (2018) | Longitudinal | Mental health-5 (SF-36) | Yes+ (C) | |
| Kroesen and De Vos (2020) | Longitudinal | Mental healthI-5 | Yes+ (RC) | NA |
| Page and Nilsson (2017) | Quasi-experimental | GHQ-12 | Yes+ (C) | |
| Jacob et al. (2021) | Quasi-experimental | MCS-SF12 | Yes+ | |
| *Mental health variable: quality of life* | | | | |
| Cerin et al. (2009) | CS | SF-12 | No | Leisure PA amount/wk |
| Jurakić et al. (2010) | CS | SF36 | Yes- | NA |
| Pucci et al. (2012) | CS | WHOQoL-BREF | Yes+ (M) | Leisure walking (W) |
| Gómez et al. (2013) | CS | SF-8 | Yes- (W) | Leisure time PA |
| Scarabottolo et al. (2019) | CS | SF-36 | No | |
| Cobbold et al. (2022) | CS | WHOQoL-BREF | Yes+ | NA |
| Scarabottolo et al. (2022) | Longitudinal | SF-36 | Yes+ | NA |
| de Geus et al. (2008) | Non-randomised CT | SF-36 | Yes+ (C) | None |
| Mutrie et al. (2000) | RCT | SF-36 | Yes+ | NA |
| Mutrie et al. (2002) | RCT | SF-36 | Yes+ (W) | NA |

(*Continued*)

**Table 3.** (*Continued*)

| References | Study design | Mental health measure | Evidence of relationship | Largest covariate |
|---|---|---|---|---|
| Baker et al. (2008) | RCT | EQ-5D | No (W) | |
| Neumeier et al. (2020) | RCT | SF-36 | Yes+ (C) | NA |
| *Mental health variable: self-efficacy* | | | | |
| Bergland et al. (2010) | CS | Personal Mastery Scale | Yes+ (W) | Age. |
| Molina-García et al. (2010) | CS | Physical Self-Efficacy Scale | Yes+ | |
| Bélanger-Gravel et al. (2016) | CS | "How confident are you in your capability of using BIXI if you chose to do so?" | Yes+ (C) | NA |
| Brainard et al. (2019) | CS | "To what extent do you agree with the statement: I can achieve most of the goals I set myself?" The scale was from (1) Strongly agree to (5) Strongly disagree | Yes+ (W) | |
| *Mental health variable: self-reported health* | | | | |
| Rasciute and Downward (2010) | CS | "How is your health in general?" (very good, good, fair, bad, very bad) | Yes+ | |
| Bergland et al. (2010) | CS | "How do you judge your own general state of health?" | No (W) | Age |
| Hansson et al. (2011) | CS | "How do you feel right now, physically and psychologically, considering your health and your well-being?" + sickness absence days | No (2) | |
| Mason et al. (2016) | CS | PCS-SF12 | No | |
| Mason et al. (2016) | CS | "In general, would you say your health is good, very good, excellent?" | Yes+ | |
| Sener and Lee (2017) | CS | 1-item How would you rate your health? | No | |
| Mattisson et al. (2018) | CS | Long-term illness + Walking difficulties + BMI | Yes+ | |
| Clark et al. (2020) | CS | "In general, would you say your health is? (poor to excellent)" | Yes+ (C) | |
| Muñiz et al. (2021) | CS | "In the last twelve months, would you say your health has been very good, good, fair, bad, or very bad?" | Yes+ | |
| Mytton et al. (2016) | Longitudinal | PCS-8 (from SF-8) + sickness absence days | Yes+ (C) | |
| Avila-Palencia et al. (2018) | Longitudinal | "In general, how would you say your health is?" (SF-36) | Yes+ | |
| Jacob et al. (2021) | Quasi-experimental | PCS-SF12 | Yes+ | |
| *Mental health variable: sleep* | | | | |
| Hansson et al. (2011) | CS | Perceived sleep quality: "Do you think you get enough sleep to feel rested?" | No | |
| Foley et al. (2018) | CS | Sleep duration (minutes/day) | Yes- | Leisure moderate-vigorous PA |
| Vancampfort et al. (2018) | CS | "Overall, in the last 30 days, how much of a problem did you have with sleeping, such as falling asleep, waking up frequently during the night or waking up too early in the morning?" | Yes+ | |
| *Mental health variable: social support* | | | | |
| Smith (2012) | CS | Daily Life Interview | Yes+ | |
| Avila-Palencia et al. (2018) | Longitudinal | "How often do you have contact with your friends and/or family?" | Yes+ (W) | |
| *Mental health variable: stress* | | | | |
| Hansson et al. (2011) | CS | "Do you feel stressed in your everyday life?" | No (2) | NA |
| Avila-Palencia et al. (2017) | CS | PSS4 | Yes- (C) | Sex (women higher) |
| Mattisson et al. (2018) | CS | "Do you feel stressed in your everyday life?" | Yes- | Trip distance |
| Sattler et al. (2020) | CS | PSS + PSQ (7 statements) | Yes- | Commuting time |
| Sattler et al. (2020) | CS | PSS + PSQ (7 statements) | No | Age, commuting time |

**Table 3.** (*Continued*)

| References | Study design | Mental health measure | Evidence of relationship | Largest covariate |
|---|---|---|---|---|
| Avila-Palencia et al. (2018) | Longitudinal | PSS-4 | Yes- (C) | |
| *Mental health variable: travel satisfaction* | | | | |
| St-Louis et al. (2014) | CS | Statements related to aspects of satisfaction with a given mode | Yes+ | |
| Friman et al. (2017) | CS | STS | Yes+ | Age |
| Smith (2017) | CS | Commute Well-being Scale | Yes+ | Congested (car), crowded transit |
| Singleton (2019) | CS | STS | Yes+ | Travel usefulness |
| Fan et al. (2021) | CS | STS | Yes+ | Residential environment |
| Liu et al. (2021) | CS | STS | Yes+ | |
| *Mental health variable: vitality/exhaustion* | | | | |
| Hansson et al. (2011) | CS | Vitality Scale (SF-36) | No (2) | |
| Mattisson et al. (2018) | CS | Vitality Scale (SF-36) | No | Trip distance |
| Clark et al. (2020) | CS | STRAIN "Have you recently felt constantly under strain?" "last few weeks" | Yes+ (W) | |
| Avila-Palencia et al. (2018) | Longitudinal | Vitality scale (SF-36) | Yes+ | NA |

*Note:* (1) only younger adults; (2) no significant relationship – healthy commuter effect. Study design – CS, cross-sectional; RCT, randomised controlled trial. Mental health variable – CES-D, Centre for Epidemiologic Studies Depression Scale; GHQ, general health questionnaire; HSCL, Hopkins symptoms checklist; MCS, mental component score; PANAS, positive and negative affect scale; PCS, physical component score; PSQ, perceived stress questionnaire; PSS, perceived stress scale; SF, short form; STS, satisfaction with travel scale; WEMWBS, Warwick-Edinburgh mental well-being scale; WHOQoL-BREF, World Health Organization quality of life short version. Evidence of relationship – C, cycling; RC, reverse causality; W, walking. Largest covariate – PA, physical activity.

showed that active travel significantly impacts mental health and not vice versa, so this bias might not be as significant for the outcomes of interest here.

## Quality of life

Most reviewed studies, particularly experimental designs, relate to Health-Related Quality of Life (HRQoL). On one side, this evidence seems to approach the outcomes by clearly distinguishing the purpose of physical activity (work/occupational, leisure time, transportation, sport and exercise). Conversely, Karimi and Brazier (2016) note that health, HRQoL and quality of life have been wrongly employed and confused (e.g., HRQoL tools measuring self-perceived health). In this review, we differentiate between the studies that used the Short Form Health Survey (SF-36) (Ware and Sherbourne, 1992) and all its versions to measure the *quality of life.* Mutrie et al. (2000) conducted the first inconclusive randomised controlled trial (RCT) and followed up with a similar study 2 years later (Mutrie et al., 2002). The findings confirmed a positive association, although significant only for walking (not cycling) as a means of transport. Later, the positive association was also reported by a non-randomised controlled trial using the same instrument (SF-36), but this time results were significant only for cycling and not for walking (De Geus et al., 2008). That same year, another RCT, using the Euroqol EQ-5D (The EuroQoL Group, 1990), claimed no correlation between active travel as walking (steps/day) and quality of life (Baker et al., 2008). More recently, an RCT (Neumeier et al., 2020) and a longitudinal study (Scarabottolo et al., 2022) have found a positive relationship between the SF-36 scores and active travel modes. A cross-sectional study (85) recently showed a positive correlation through the WHOQoL-BREF (The Whoqol Group, 1998; Murphy et al., 2000). However, the same tool in a different country (Fleck et al.,

2000) found that active travel benefits were significant only for the physical domains (e.g., body pain, functional capacity) and only in men (Pucci et al., 2012). Overall, cross-sectional studies have produced mixed results. Four reported a negative (Jurakić et al., 2010; Gómez et al., 2013) or a non-significant association (Cerin et al., 2009; Scarabottolo et al., 2019) using a version of the SF-36.

## Affect

The 11 studies identified used 11 measures to operationalise the outcome *affect*. Findings based on cross-sectional designs are mixed. Two reported a positive correlation between active travel and positive affect (Gatersleben and Uzzell, 2007; Lira and Paez, 2021). Another confirmed these results only for walking (Rasciute and Downward, 2010) and two more for cycling (Lancée et al., 2017; Kaplan et al., 2019). We also found cross-sectional evidence of a negative (Office for National Statistics, 2014) and non-significant (Morris and Guerra, 2015; Brainard et al., 2019) relationship between travelling actively and being happy. The experimental literature refers to an RCT measuring active travel as walking (steps/day) (Baker et al., 2008) and a quasi-experimental study investigating e-bike use (Page and Nilsson, 2017). They used PANAS (Watson et al., 1988) and diary analysis to measure affect. Despite the different methodologies, both reported an increase in positive affect in active travellers and not passive ones. The only longitudinal study we found confirmed this result by measuring active travel with the smartphone app Daynamica and affect on the Travel Mood Scale (Glasgow et al., 2018, 2019).

## Eudaimonia

Although the five studies identified aimed to investigate eudaimonia as the "sense of purpose and meaning in life", they all used different constructs and instruments, such as structured

questionnaires (e.g., Flourishing Scale) or ad hoc statements. One cross-sectional study highlighted a significant negative relationship (Office for National Statistics, 2014), and another reported the same result only for cycling (Kaplan et al., 2019). Two more cross-sectional designs showed a non-significant relationship (Brainard et al., 2019; Liu et al., 2021). A quasi-experimental analysis confirmed this result; however, their intervention was based on AM only as e-bike use (Page and Nilsson, 2017). The most recent study (Liu et al., 2021) demonstrated that AM and eudaimonia are *indirectly* related since both significantly correlate with hedonic well-being (positive and negative affect and life satisfaction).

### Life satisfaction
This outcome has been investigated using single-item instruments like, "Overall, how satisfied are you with your life?" One cross-sectional study showed that people actively travelling were less satisfied with their life (Office for National Statistics, 2014). Another confirmed this result but only for people younger than 50 (Sener and Lee, 2017). Two more reported the opposite, so active travel was positively related to life satisfaction (McCarthy and Habib, 2018; Wang et al., 2021), in one case significant only for walking (Chng et al., 2016). Two other analyses recorded a non-significant relationship between the factors (Brainard et al., 2019; Clark et al., 2020). Together these results suggest that the context in which active travel is chosen shapes how it is perceived. The only (quasi-)experimental evidence shows that changing from public transport to active travel improved people's life satisfaction (Jacob et al., 2021).

### Travel satisfaction
The search found six cross-sectional studies concerning travel satisfaction and active commuting. The results are consistent and encouraging. While the oldest article used specific *aspects of satisfaction* (St-Louis et al., 2014) to operationalise travel satisfaction, the other five studies used some variant of the Satisfaction with Travel Scale (Friman et al., 2017; Smith, 2017; Singleton, 2019; Fan et al., 2021; Liu et al., 2021). These studies showed a significant positive correlation between walking and cycling and travel satisfaction.

### Stress
Four cross-sectional studies explored the link between transport modes and stress levels. The oldest reported a non-significant association between active travelling and stress, claiming the "healthy commuter effect" mentioned above (Hansson et al., 2011). The same measure (a single-item "Do you feel stressed in your everyday life?") was used by a later cross-sectional study (Mattisson et al., 2018), while two more (Avila-Palencia et al., 2017; Sattler et al., 2020) using a structured questionnaire (Perceived Stress Scale, PSS-4) (Cohen et al., 1983) to reveal an inverse relationship between active travel and stress levels. Finally, Avila-Palencia et al. (2018) obtained the same result, although significant only for cycling, through a longitudinal study design using the PSS-4.

### Depressive symptoms
Although the present scoping review includes only empirical studies, the results of a recent review investigating the relationship between active commuting and depression among adults (Marques et al., 2020) are worth mentioning. Firstly, no experimental evidence was found. Five of the seven identified studies reported no significant relationship. The one with a longitudinal design reported decreased depressive symptoms after changing

from inactive to active mode (Knott et al., 2018). We identified two cross-sectional studies not included in this review. Bergland et al. (2010) studied *outdoor walking ability* or outdoor mobility and found no significant relationship with depression levels. However, Muñiz and others lately reported a negative relationship between these variables (Muñiz et al., 2021). In addition, a longitudinal simulation by Yang et al. (2019) using agent-based modelling (ABM) (Nianogo and Arah, 2015) predicted almost no effect of walking on depression prevalence. Each study measured the outcome by applying different approaches.

### Anxiety
One cross-sectional study considered only walking as the active travel mode and measured anxiety using the Hopkins Symptoms Check-List (HSCL) (Parloff et al., 1954); it found no significant relationship (Bergland et al., 2010). In 2014, the ONS (UK) analysed the data from a survey asking, "Overall, how anxious did you feel yesterday?" (Office for National Statistics, 2014). Commuters were found to be more anxious than non-commuters, and those who actively travelled to places, especially for longer than 15 min, reported worse anxiety levels. On the other hand, a similar but more recent analysis found no association between anxiety levels and active commuting (Brainard et al., 2019).

### Social support and loneliness
The literature about social support and active travel is minimal and focused on older adults. Two studies explored social contact with friends and family and loneliness levels. They use the same instrument (UCLA Loneliness Scale; Version 3) (Austin, 1983; Russell, 1996). A cross-sectional design examined the mobility of a sample of older adults (74–98 years old) and found that "disrupted engagement with others" due to reduced mobility (stopped driving a car and the inadequacy of transportation) was associated with fewer current relationships which also translated in higher loneliness feelings (Smith, 2012). The participants stated that "reaching others" would be their primary strategy to recover those contacts. The longitudinal study (Avila-Palencia et al., 2018) considered unimodal and multimodal travelling (e.g., public transport and walking) and confirmed these findings. Incorporating walking into multimodal trips was associated with higher social contacts, and cycling in multimodal trips was related to fewer feelings of loneliness.

### Self-efficacy
All included studies were cross-sectional and typically assumed that self-efficacy was a determinant of transport mode choices, not a result. Two papers studied outdoor mobility in the elderly and active travel in adults linked to self-efficacy using structured scales (Bergland et al., 2010; Molina-García et al., 2010). Cross-sectional evidence was also gained from implementing a public bicycle share program (BIXI – Bicycle-taXI, Canada) designed to increase people's intention and self-efficacy to use the bike as a means of transport (Bélanger-Gravel et al., 2016). They measured self-efficacy with ad hoc statements, and the program led to an increase in active travel and self-efficacy. The most recent evidence (Brainard et al., 2019) showed a significant positive relationship only for walking. Overall, the results are modest and hard to interpret.

### Sleep
Evidence regarding sleep patterns and AM is limited to cross-sectional studies. One found no significant results (Hansson

et al., 2011). Two were published in 2018 and used data from public health surveys but operationalised sleep behaviour differently. For example, Foley et al. (2018) used self-reported sleeping time (minutes/day) records, while Vancampfort et al. (2018) assessed sleep through a single item about sleeping issues (falling asleep, waking up frequently or too early in the morning). The former found that active travellers reported shorter sleep duration, yet, the latter found that people actively travelling scored better sleep.

### Vitality/exhaustion

The evidence about this outcome is mainly cross-sectional. Avila-Palencia et al. (2018) identified a significant positive relationship between cycling and walking as a means of transport and vitality scores. Clark et al. (2020) claimed that walking to work was associated with reduced strain, thus, better vitality. Nevertheless, low vitality (exhaustion) has shown no significant relationship with active travel (Hansson et al., 2011; Mattisson et al., 2018). The inconsistency might likely be due to the methodology and construct definition differences.

### Self-reported health

Self-reported health has mainly been investigated via single-item questions like "How is your health in general?" (Rasciute and Downward, 2010) or "How do you judge your general state of health?" (Bergland et al., 2010). In other cases, authors used physical health scales, such as the Physical Component Score of the SF-36 (PCS). Researchers have also measured people's health status as the sum of outcomes such as sickness and absence from work (Hansson et al., 2011; Mytton et al., 2016), long-term illness, walking difficulties and obesity (BMI) (Mattisson et al., 2018). Considering the established beneficial influence of transport physical activity on physical health (Marques et al., 2020; Alattar et al., 2021), it was no surprise that active travel modes improved objective and subjective health measures. Objectively, walking and cycling reduced sickness absence days, illness levels, walking difficulties and obesity scores (Mattisson et al., 2018). Subjectively, cycling and walking were associated with better self-perceived health compared to car driving or taking the bus (Avila-Palencia et al., 2018; Clark et al., 2020). These studies, investigating mental and physical aspects of health, confirmed the positive relationship between these components of overall health (Rasciute and Downward, 2010; Mason et al., 2016; Sener and Lee, 2017; Mattisson et al., 2018). Five cross-sectional designs claimed a positive relationship between AM and self-reported health, four using specific questions (Rasciute and Downward, 2010; Humphreys et al., 2013; Mason et al., 2016; Mattisson et al., 2018; Muñiz et al., 2021). Two more confirmed these results: walking (Bergland et al., 2010) and cycling (Clark et al., 2020). Other cross-sectional findings described a non-significant relationship (Bergland et al., 2010; Hansson et al., 2011; Mason et al., 2016; Sener and Lee, 2017).

The longitudinal and quasi-experimental evidence indicated a positive relationship between better self-reported health and walking and cycling (Avila-Palencia et al., 2018), maintaining cycling over time (Mytton et al., 2016), and changing from passive to active transport modes. In conclusion, self-reported health appears to have a stronger significant relationship with active travel compared to mental health outcomes, such as self-efficacy, self-esteem, depression and anxiety (Bergland et al., 2010), stress and vitality (Mattisson et al., 2018) and overall mental health (Hansson et al., 2011) and mental well-being (Humphreys et al., 2013).

## Discussion

To our knowledge, this review is the first to examine a vast range of mental health outcomes related to "AM" in healthy adult populations. Moreover, it highlights promising evidence of the positive effects that AM might have on mental health, both in terms of prevention and interventions to reduce symptoms.

In this section, we attempt to summarise areas where methodologies and outcomes seem to agree and where there is a divergence or a lack of relevant investigations.

### Key findings for mental health outcomes

Regarding the number of studies, mental health, quality of life, and the affective component of subjective well-being have received the most extensive attention. These, followed by self-reported health, also appeared as the most promising regarding a favourable relationship with active travel modes, confirmed by the experimental or quasi-experimental investigations. Although evidence regarding travel satisfaction, life satisfaction, stress, loneliness and social support, and self-efficacy was limited, results were encouraging from a health promotion perspective. Eudaimonia, depressive symptoms, anxiety, sleep, and vitality had limited or mixed findings. No studies about self-esteem and resilience could be found.

### Points of consensus and gaps in the literature

We grouped these into five areas.

### What is active mobility?

Given the differing backgrounds and study designs, a consensus definition is complex. This problem is discussed further in section "AM, the exposure factor". Study differences must be interpreted based on the included and excluded behaviours. For example, public transport represents a contentious factor since some studies implicitly discount the walking associated with travel to bus stops or train stations (Neumeier et al., 2020). Furthermore, the trip purpose or domain might be crucial. Studies have reported that leisure-time walking and cycling consistently correlate with the quality of life, whereas transport physical activity does not (Rasciute and Downward, 2010; Pucci et al., 2012). Finally, some authors explicitly investigated walking or cycling (Mutrie et al., 2002; De Geus et al., 2008); others found significant results only for one active travel modality and not the other, without further exploration to explain these findings (Mytton et al., 2016; Avila-Palencia et al., 2018).

### Does the quantity matter?

Not only mode and purpose but also duration and distance could impact the beneficial health effects of AM (Singleton, 2019). As a form of physical activity, frequency, duration, and intensity might mediate or moderate AM's health benefits (Ekkekakis et al., 2011; Chan et al., 2019). While it is not yet clear if a "dose–response" relationship exists in the results, authors have found that for each additional commuting minute, especially for trips longer than 10–15 min, people using active travel modes (and public transport) tend to report lower eudaimonia, life satisfaction, travel satisfaction, happiness, and overall mental health and higher strain and anxiety feelings (Office for National Statistics, 2014; St-Louis et al., 2014; Singleton, 2019; Clark et al., 2020; Wang et al., 2021). Recently, a study on travel satisfaction confirmed these results (De Vos et al., 2022). Compared to commute mode, commute duration showed a

stronger association with the outcome and appeared to moderate the relationship with the exposure. While walking and cycling usually refer to shorter trips (less than 20 min), public transport is used for more than 30 min commutes. This time frames likely explain the resulting satisfaction levels.

Still, commuting time did not relate to the quality of life (Neumeier et al., 2020), eudaimonia (Page and Nilsson, 2017) and life satisfaction (Clark et al., 2020), as reported by experimental and quasi-experimental evidence and more recent cross-sectional designs.

### Does the context matter (where and when)?
Scientists have demonstrated that walking through green and blue areas (natural environments) (Glasgow et al., 2019) provides advantages in terms of positive mood and overall mental health (Zijlema et al., 2018; Wicks et al., 2022). However, when investigating commuting, authors rarely specify where the active travel occurred or within what type of settings. The surrounding environment might represent an obstacle or an advantage to the beneficial effect of walking and cycling on mental health, well-being and mood (Glasgow et al., 2019). Opposite to the consistent positive effect on well-being shown by walking, cycling's benefits could be influenced by "disutility" connected to environmental barriers (Rasciute and Downward, 2010). Public bicycle stations around the home/work/study addresses, levels of greenness and "bikeability/walkability", and the presence and length of sidewalks are all examples of external characteristics influencing active travel behaviour (Avila-Palencia et al., 2017; Glasgow et al., 2019).

Regardless of the transport mode, contextual characteristics might directly impact stress levels (Avila-Palencia et al., 2017). The season or the hour of the day could also be considered "objective external characteristics". A study in Canada observed that people tend to use active commuting in spring/summer and switch to "public transit" in winter. Also, those actively travelling in harsh weather conditions showed significantly lower travel satisfaction. Lastly, they noticed that people working during regular hours were more satisfied than those who travelled to work during irregular hours, possibly due to a lack of adequate public services during those hours (St-Louis et al., 2014).

### Do personal characteristics matter?
Individual and demographic characteristics are potentially confounding variables in the link between AM and mental health. However, there is no consensus as to which are most influential. Such characteristics include sex, age, education, employment status, monthly income, ethnicity, marital/relationship status, household (cohabiting status; the number of children), and baseline health (operationalised as weight, BMI, and physical activity levels).

Age and sex have significantly impacted the use of AM modes and their relationship with health benefits. Younger people, particularly men, travel more frequently by walking and cycling (Bergland et al., 2010; Scarabottolo et al., 2022), albeit not in all cases (Cobbold et al., 2022). Age also appeared to moderate active travel effects on life satisfaction, emotional well-being and travel satisfaction, so people older than 50 opting for active travel had poorer mental health scores (Friman et al., 2017; Sener and Lee, 2017). Regarding differences between men and women, the results are equivocal. Studies have shown that commuting on foot or by bike significantly benefits mental health and quality of life only in men (Ohta et al., 2007; Pucci et al., 2012). However, little experimental and quasi-experimental evidence claimed that changing from passive to active travelling improves mental health in both

men and women (higher vitality and overall mental health), but physical health significantly increased in women and decreased in men (De Geus et al., 2008; Jacob et al., 2021). Also, changing from active to passive modes meant poorer physical health in both sexes but better mental health in women (Jacob et al., 2021). Other than age and sex, social disparities might have a role in influencing quality-of-life improvements due to AM (Gómez et al., 2013). For instance, it was noted that in low-income countries, people might be forced to walk or cycle as a means of transport because of their poverty conditions and associated feelings of discomfort (Pucci et al., 2012; Lira and Paez, 2021). Nevertheless, the moderating role of these factors remains unclear and only potential since the results of their influence have not been widely confirmed (Sattler et al., 2020).

### Does study methodology matter?
Most studies reviewed were cross-sectional, and many used national population survey data. However, it is difficult to identify any element of the study design on which consensus was achieved. Indeed, different kinds of studies were considered suitable for examining different aspects of the question. Still, the only experimental evidence identified dates back almost 20 years ago. The interest in this phenomenon spans the fields of environmental studies, transportation studies, economics, public health and exercise science. Thus, the instruments and terminology employed, as valid and reliable as they may be, are frequently not very specific, which can create concerns in comparing the results. Furthermore, most information about active commuting and mental health was self-reported. However, authors have demonstrated that self-report and objective measures might have the same validity in reporting AM characteristics (Gebel et al., 2009; Laeremans et al., 2017). Focusing on mental health and quality of life outcomes in adults resulted in neglecting other vital aspects of adult people's health like cognitive and neurological function, the former considered predominantly in school-aged populations. Together with self-efficacy, and social support, self-esteem and resilience are considered mental health "protective" factors (Keyes, 2002, 2005, 2007). *Resilience* is widely accepted as a vital psychological aspect of healthy functioning from an individual and public health perspective (Kelly et al., 2018). Despite its significance, active travel research has wholly ignored these aspects.

Despite well-validated instruments to measure specific mental health outcomes exist, their absence in the current literature is a lacuna that future studies in this field should address.

### Limitations of the present study

The search method used by this review to identify studies was tightly focused on English language studies with AM as the exposure and mental health as the outcomes and healthy adults as the sample, so important literature may have been excluded where, for example, the distinction between walking and cycling as a means of transport and as a leisure-time activity was not completely clarified, where the population was younger or older, or the reports were written in other languages.

In order to limit the scope of an already complex review, we have not attempted to untangle the possibly contrasting contributions of mixed-mode travel to the outcomes of interest and have avoided studying the effects of partially active modes, including electric bicycles and scooters.

The studies included in this review used a variety of research designs and methodologies. Most employed cross-sectional

analyses, precluding causal inferences about the relationship between AM and mental health.

Most included studies focused on single ethnical-cultural-social populations (USA, Belgium, New Zealand, Japan, older adults, or workers), meaning their results are only partially comparable and might not be valid for other populations.

Lastly, we did not attempt to account for publication bias due to the absence of unpublished studies with no or incoherent conclusions.

### Future perspective: A unified conceptual and empirical framework

Based on a review of the current state of the literature, we believe that more prospective experimental studies are needed, using specific instruments to operationalise mental health outcomes. For example, the *healthy commuter effect* discussed above might be crucial to investigate through the experimental approach, producing new AM unbiased research. In line with the health promotion and precision health approach (Ryan et al., 2021), it is necessary to understand what factors might change AM's advantageous effect on people's health. To this end, it would be helpful to better define AM as a physical activity (in frequency, intensity and duration) that would allow a more objective comparison of different modes, such as walking, cycling, e-bike use or even wheelchair use and skateboarding. Additionally, critical environmental factors, such as air pollution or green areas, are more accessible information nowadays using lightweight, portable instruments to track people's transport routes. However, there remain many open questions. To what extent is journey purpose influential? Are health benefits due to an additive or synergic relationship between moderators and exposure? Are there groups of people who are more susceptible? Active travel has a potentially significant and bidirectional relationship with social support and loneliness that has been little investigated. Does active travel increase social connection and reduces loneliness? We suggest a unified and shared framework with precise terminology, properly operationalised variables and specific measurement tools to assist a comparative evaluation of the studies already published and the design of new research in this area. Although a similar task has been attempted recently (Götschi et al., 2017), the authors aimed to identify the determinants of AM behaviours more than their health benefits and mental health was scarcely considered. Still, the current literature is highly mixed.

Taken at face value, the results presented in this study suggest that in many cases different cohorts respond differently to the experience of AM; policymakers should carefully consider the intended targets of any intervention as responses may not be uniformly positive. Specific recommendations can only be given when solid evidence has been gathered to answer the questions posed above.

### Conclusions

This review lists outcomes related to mental health that have received scientific attention when studying AM's effects on health. Although limited and mainly cross-sectional, this evidence has suggested that AM, defined as walking and cycling to get to and from places, provides significant benefits in terms of mental health. A few experimental designs have mainly investigated the quality of life. Moreover, since the millennium, authors have broadened their interest in examining overall mental health, positive and negative affect and self-reported health. More recently, cross-sectional studies have also focused on other important individual and public health outcomes like stress, life and travel satisfaction, loneliness and social support. Also, depressive and anxiety symptoms and sleep quality have been surprisingly neglected, considering their global health-related burden at an individual and societal level. Moreover, eudaimonia, self-efficacy, self-esteem, and resilience, which might all have a role in this subject, are barely considered. Finally, as findings in different populations are hardly comparable, we described the approaches used in the studies included. We hypothesise that active travel modes and mental health benefits have been explored from a non-strictly psychological scientific perspective, leading to inconsistencies in methodologies, outcome definitions and instruments and omitting essential health factors.

We hope this review will motivate new and diverse approaches to research how AM and mental health interact, aiming to promote sustainable lifestyles and improve public health and personal well-being.

**Open peer review.** To view the open peer review materials for this article, please visit http://doi.org/10.1017/gmh.2023.74.

**Supplementary material.** The supplementary material for this article can be found at https://doi.org/10.1017/gmh.2023.74.

**Data availability statement.** The authors confirm that the data supporting the findings of this study are available within the article (and/or its Supplementary Materials).

**Author contribution.** The authors equally contributed to the conceptualisation of the study. L.S. and S.M.M. conceived the scoping review design. L.S. and D.N.M. planned the methodology. L.S. conducted data acquisition and data extraction, and D.N.M. completed it with additional numerical values. L.S. performed narrative analysis, interpretation and reporting. D.N.M. performed the numerical analyses and graphical illustration. D.N.M. and A.T. revised the drafts. Finally, D.N.M. and L.S. edited the interim versions of this manuscript. All authors have read and agreed to the published version of the manuscript.

**Financial support.** This research received no specific grant from any funding agency in the public, commercial or not-for-profit sectors.

**Competing interest.** The authors declare no conflict of interest.

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
