## [Reviewer Report]

31/05/2023

To the attention of Global Mental Health Editor-in-Chief

Dear Sir/Madam,

We wish to submit a research manuscript entitled “Active Mobility and Mental Health: A Scoping Review towards a Healthier and Greener World” for publishing in your esteemed journal.

First of its kind, this comprehensive review examines the existing literature on active transportation modes' benefits in terms of mental health outcomes. The extended time range and the number of variables covered explain the extensive length of this work. 

From the perspective of health promotion and sustainable development, we selected the Global Mental Health journal because the literature includes many countries worldwide and because the potential impact of the present paper wishes to support international and interdisciplinary collaborations.

All of this manuscript’s authors (Luana Scrivano, Alessia Tessari, Samuele Maria Marcora, and David Neil Manners) agreed to submit the manuscript to the journal Global Mental Health. We also decided to transfer the copyright from the authors to the journal. The manuscript has been prepared per the journal’s guidelines and checked for language correction. 

We confirm this work is original, and the manuscript is not currently considered for publication elsewhere.

Please address all the correspondence on this manuscript to me at luana.scrivano2@unibo.it.

Thanking you

Sincerely

Luana Scrivano

---

## [Reviewer Report]

Overview:

On the whole I found this coherent and a good literature overview, structured well enough for people who are only interested in one sub topic to find the information they wanted, although there are plenty of minor issues and one puzzle.

I don’t know how novel this is, but it does a good job of being transparent and rigorous and a decent overview. Also strongly agree with conclusions, that RCTs would be useful and more definitive. The 4 figures in main text are interesting. There is a problem with Figures 4 (or 5?) – 10, which is my only major comment.

MAJOR

I don’t see Figures 4-10, as described in this sentence

“Finally, the implicit effect sizes of statistically assessed outcomes are shown in Figures 4 to 10.”

Figure 4 seem to be something else. Figures 5-10 aren’t discussed again. So that all is confusing.

MINOR:

Progressing in abstract should be Progress. I also think a better word might be ‘improvement’ or simply ‘links to’

Impacts, “related” on pg 3 line 15: should be ‘relevant’

There are minor issues like missing spaces between narrative text and references(like so)

I appreciate the clear description of how the search phrases were developed.

“The extracted were used to describe characteristics”

Needs a noun, maybe should be

The extracted data were used to describe characteristics

/or/

The extractions were used to describe characteristics

cross-sectional case-control studies : these are 2 different study designs. Should that be “cross-sectional or case-control studies”

similarly, should put a comma to make phrase

(non-randomized controlled, cohort or longitudinal follow-up studies)

Is data plural or singular in this article? Line 55 pg 6

The albatross plots were an interesting way to visualise the findings. On balance I think I liked them, showed difference between RCTs & X sectional quite effectively.

Eudaimonia is an uncommon word in English; would help to define it before page 10, at 1st use ideally. It gets defined again a few more times after pg 10, probably not all necessary.

I’m not sure the authors did enough to explain that people who use public transport inherently walk more; to get to and from the bus. Indeed, having a bus pass is a good predictor of higher physical activity. That’s why public transport use gets mixed up with calculating time spent in active mobility. Increasingly popular Use of hire electric scooters & bicycles is going to complicate the picture, too.

I wouldn’t be as confident as authors about the cause and effect relationship of AT-AC-AM & good mental health; maybe the relationship runs the other way, good MH helps make AT-AC-AM easier. We need more RCTs, especially some more recent ones!

OTHER

Someone is doing an IPD meta-analysis (American PhD student, I think) about active travel and health outcomes, you may want to check syst-review protocol registries to see what they are doing, their progress.

Could incorporate wider environmental and social benefits of more people not moving around urban environments in high pollution vehicles, and lower risk of traffic accidents (although that may not be proven, and is complicated by things like high speed rental electric scooters).

Age-sex relationship: may be changing. What people did in past (social norms then) may not predict patterns in future; long distance female cyclists have become more common, for instance.

---

## [Reviewer Report]

PREVIEW

Active Mobility and Mental Health: A Scoping Review towards a Healthier and Greener World

SUMMARY

The topic of this scoping review is highly relevant and needed. I’m sure once it will be published, a lot of researchers would use it as a starting point to define future research directions. So, really happy to see this!

My main concerns are the over-promising Introduction, the lack of specificity in Methods, and a missing deeper Discussion in limitations and implications of the review. Find some suggestions in my comments below.

GENERAL COMMENTS

- Title & Introduction: the list of aims seems really ambitious and I’m not sure all of them are achieved (e.g., I haven’t seen any framework as a result of this review). The same happens with the second part of the title (‘A Scoping Review towards a Healthier and Greener World’). The review is focused on mental health and well-being and does not have that much emphasis on green space (assuming that the word ‘greener’ in the title refers to green space). I think the authors would guide better the readers avoiding over-promises in the title and the introduction of the paper. I suggest to re-write the title and the aims part of the Introduction being a bit more accurate and aligned with what is actually presented in the paper.

- Search strategy:

o it would be good to know the date when each source was last searched or consulted and if any filters and limits were used in the search.

o Could the authors be a bit more specific in the methods they used to decide whether a study met the inclusion criteria? Did they used a form? The text says “The same author (LS) conducted the full-text assessment, and the inclusion of potential studies was agreed upon by consensus with the others (DNM, AT, SMM).” If I’m understanding this properly, only one person read the papers, how the agreement about the inclusion of studies was done then?

- Study quality and risk of bias: the authors mention the ‘Oxford Centre for Evidence-Based Medicine classification’ but it is not very clear to me how this classification was used. I suggest to re-write the paragraph being more specific and clearer about if this classification was the actual tool used to assess the quality and risk of bias and if so, how it was used (e.g. it was used by one reviewer, there was any agreement process with other reviewers, etc).

- Descriptive analytical synthesis: it would be good if the authors could extend a bit more on the explanation of the ‘descriptive-analytical’ method for those readers who are not familiar with this term. Any literature reference would be helpful too.

- Mapping of terminology: it seems the authors decided to adopt ‘active mobility’ as the term they prefer to use. I suggest including a statement somewhere in the text, maybe this section, in which they explain their election and why they made it.

- Limitations: I miss a deeper discussion of the limitations of the evidence included in the review. What are the implications of the limitations identified (e.g., Mostly cross-sectional studies, self-reported methods)? The authors also need to discuss limitations of the review processes used.

- Future perspective: this section mainly discusses implications for future research. Any implication of the results for practice and policy?

SPECIFIC COMMENTS

- Appendix B uses a lot of acronyms in titles (e.g. MH, AM). They might seem self-explanatory, but I would recommend using the full word(s). If the authors prefer not to, make sure the acronyms are properly detailed in the document.

---

## [Reviewer Report]

31/05/2023

To the attention of Global Mental Health Editor-in-Chief

Dear Sir/Madam,

We wish to submit a research manuscript entitled “Active Mobility and Mental Health: A Scoping Review towards a Healthier and Greener World” for publishing in your esteemed journal.

First of its kind, this comprehensive review examines the existing literature on active transportation modes' benefits in terms of mental health outcomes. The extended time range and the number of variables covered explain the extensive length of this work. 

From the perspective of health promotion and sustainable development, we selected the Global Mental Health journal because the literature includes many countries worldwide and because the potential impact of the present paper wishes to support international and interdisciplinary collaborations.

We declare that all this manuscript’s authors (Luana Scrivano, Alessia Tessari, Samuele Maria Marcora, and David Neil Manners) agreed to submit the manuscript to the journal Global Mental Health. We also agreed to transfer the copyright from the authors to the journal. The manuscript has been prepared per the journal’s guidelines and checked for language correction. 

We confirm this work is original, and the manuscript is not currently considered for publication elsewhere.

Please address all the correspondence on this manuscript to me at luana.scrivano2@unibo.it.

Thanking you

Sincerely

Luana Scrivano

---

## [Reviewer Report]

I am pretty happy with how the authors have addressed my comments. Willing to see it published! All the best.